# Precise Identification of Chromosome Constitution and Rearrangements in Wheat–*Thinopyrum intermedium* Derivatives by ND-FISH and Oligo-FISH Painting

**DOI:** 10.3390/plants11162109

**Published:** 2022-08-13

**Authors:** Zhihui Yu, Hongjin Wang, Ennian Yang, Guangrong Li, Zujun Yang

**Affiliations:** 1School of Life Science and Technology, University of Electronic Science and Technology of China, Chengdu 610054, China; 2Crop Research Institute, Sichuan Academy of Agricultural Sciences, Chengdu 610066, China

**Keywords:** chromosome rearrangement, ND-FISH, Oligo-FISH painting, *Thinopyrum intermedium*, wheat

## Abstract

*Thinopyrum intermedium* possesses many desirable agronomic traits that make it a valuable genetic source for wheat improvement. The precise identification of individual chromosomes of allohexaploid Th. intermedium is a challenge due to its three sub-genomic constitutions with complex evolutionary ancestries. The non-denaturing fluorescent in situ hybridization (ND-FISH) using tandem-repeat oligos, including Oligo-B11 and Oligo-pDb12H, effectively distinguished the St, J and J^S^ genomes, while Oligo-FISH painting, based on seven oligonucleotide pools derived from collinear regions between barley (*Hordeum vulgare* L.) and wheat (*Triticum aestivum* L.), was able to identify each linkage group of the *Th. intermedium* chromosomes. We subsequently established the first karyotype of *Th. intermedium* with individual chromosome recognition using sequential ND-FISH and Oligo-FISH painting. The chromosome constitutions of 14 wheat–*Th. intermedium* partial amphiploids and addition lines were characterized. Distinct intergenomic chromosome rearrangements were revealed among *Th. intermedium* chromosomes in these amphiploids and addition lines. The precisely defined karyotypes of these wheat–*Th. intermedium* derived lines may be helpful for further study on chromosome evolution, chromatin introgression and wheat breeding programs.

## 1. Introduction

As a perennial grass with a native distribution throughout the Mediterranean and Eastern Europe [1], *Thinopyrum intermedium* (Host) Barkworth & D.R. Dewey has undergone the most direct selection for hay and pasture grass improvement [2]. As an allohexaploid (2n = 6x = 42), *Th. intermedium* chromosomes have been previously identified and allocated into three tentatively designated sub-genomes sets as J, J^S^ and St, where St is closely related to the genome of *Pseudoroegneria strigosa* and J is closely related to that of *Th. bessarabicum.* The J^S^ genome is a modified version of the J and St genomes [3,4]. *Th. intermedium* consists of two subspecies, namely, intermediate wheatgrass, ssp. *intermedium* and a pubescent variant ssp. *trichophorum* that carries novel and high levels of resistance to several wheat fungal diseases, such as rusts and powdery mildew, and many useful agronomic traits, including novel glutenins, and contributes to the tertiary gene pool for wheat improvement [4,5]. The production and identification of amphiploids or partial amphiploids between wheat and *Th. intermedium* is an important intermediate step for gene transfer [5]. To date, a number of wheat–*Th. intermedium* amphiploids, Otrastsyuskaya (OT), TAF46, Zhong1 to Zhong5, 78829, TAI7044, TE-3, TE253-1, TE257, TE267, TE346, SX12-787, SX12-1150 and SX12-1269, have been obtained [4,6,7,8,9,10]. The precise identification of individual *Th. intermedium* chromosomes is essential for comparative molecular and cytogenetic analysis of *Th. intermedium* species and the introgression of diverse *Th. intermedium* chromatin for breeding purposes of wheat.

Due to the complexity of the genomic composition of *Th. intermedium*, the precise recognition of individual *Th. intermedium* chromosomes in wild grass itself and wheat–*Th. intermedium* partial amphiploids is rather difficult by current molecular cytogenetic methods including C-banding, genomic in situ hybridization (GISH) and fluorescent in situ hybridization (FISH) with the reported probes [4,5,10,11,12]. Recently, we developed seven bulked pools for an oligo painting system, which enabled the assignment of the chromosomes to the Triticeae linkage groups and the characterization of wheat-alien chromosome derivatives [13]. The powerful Oligo-FISH painting methods [14], in combination with non-denaturing fluorescent in situ hybridization (ND-FISH) using multiple oligo probes [15,16,17,18,19,20], have clearly improved the efficiency and accuracy of distinguishing *Th. intermedium* chromosome segments and their introgression in a common wheat background.

In the present study, we established a standard karyotype of individual *Th. intermedium* chromosomes, which show the genomic heterogeneity of wheatgrass. Comprehensive FISH and Oligo-FISH painting enabled the precise identification of individual chromosomes and chromosome rearrangements in a wheat–*Th. intermedium* partial amphiploid and derived lines.

## 2. Results

### 2.1. Identification of Individual Thinopyrum intermedium Chromosomes

All 21 pairs of *Th. intermedium* chromosomes have been previously identified by ND-FISH using the repetitive sequences Oligo-pSc119.2 and Oligo-pTa535 as probes. Subsequently, ND-FISH using the probes Oligo-B11 and Oligo-pDb12H was performed to further characterize the mitotic metaphase chromosomes of the *Th. intermedium* accession PI440043. The Oligo-pDb12H was able to clearly classify the 14 J^S^ chromosomes of *Th. intermedium*, while the abundant signals associated with Oligo-B11 revealed the St chromosomes, and the mostly telomeric and sub-telomeric signals with Oligo-B11 on the J chromosomes [17,18]. As shown in Figure 1a, all the 42 *Th. intermedium* chromosomes were clearly distinguished into three subgenomes designated as J, St and J^S^, each with 14 chromosomes by using ND-FISH. The sequential ND-FISH patterns using probes Oligo-pSc119.2 and Oligo-pTa535, combined with the FISH patterns by Oligo-B11 and Oligo-pDb12H, were able to distinguish the individual *Th. intermedium* chromosomes in three J, St and J^S^ sub-genomes (Figure 1b).

The wheat–barley bulked oligo pools Synt1 to Synt7 were then used by sequential FISH for validation of the signal distribution and specificity for the chromosomes of *Th. intermedium* PI440043 (Figure 1). Each chromosome plate was first analyzed by ND-FISH with probe combinations of Oligo-B11 + Oligo-pDb12H (Figure 1a) and then Oligo-pSc119.2 + Oligo-pTa535 (Figure 1b,e,g,i). For example, the probe Synt5 produced strong hybrid green signals on each of three homoeologous chromosome pairs along the entire lengths of these groups of six chromosomes (Figure 1c). According to ND-FISH results, the three pairs of homoeologous chromosomes belong to each St, J and J^S^ subgenome. They are thus designated as 5St, 5J and 5J^S^ (Figure 1c). In the same metaphase cell, the probe Synt7 generated distinct red signals on three chromosomes pairs of 7St, 7J and 7J^S^ (Figure 1c), and Synt3 identified another three chromosomes pairs 3St, 3J and 3J^S^ (Figure 1d), respectively. Similarly, the bulked probes Synt1, Synt2, Synt4 and Synt6 also hybridized to mitotic chromosomes of *Th. intermedium*, and they all produced distinct signals on their individual three homoeologous chromosomes pairs (Figure 1). Therefore, all seven painting probes resulted in distinct hybridization signals covering chromosome arms along their entire lengths for each linkage group, suggesting that the bulked Oligo-based FISH painting can be used to identify each of the *Th. intermedium* chromosomes or chromosome segments of particular homoeologous groups. The 21 chromosome pairs were finally assigned to seven homoeologous groups among St, J and J^S^- subgenomes (Figure 1k).

Based on the hybridization patterns of Oligo-pSc119.2 and Oligo-pTa535 on the chromosomes of PI440043, some homologous chromosomes pairs showed different hybridization signal patterns for Oligo-pSc119.2, including 2St, 6St and 3J^S^ with the presence or absence of signals at their terminal regions. Furthermore, the chromosome pairs 7J^S^ and 5J displayed different Oligo-pTa535 signals on the pericentromeric regions (Figure 1). Importantly, the different ND-FISH patterns were visible only on one of the homologous pairs, indicating structural chromosome heterozygosity in *Th. intermedium* plants.

### 2.2. Chromosome Identification of Wheat–Th. intermedium Amphiploids

The wheat–*Th. intermedium* partial amphiploid TAF46 was selected to characterize the *Thinopyrum* chromosome compositions. Sequential ND-FISH using Oligo-pSc119.2 and Oligo-pTa535 could easily distinguish the 42 wheat chromosomes and a total of 14 *Th. intermedium* chromosomes. The hybridization signals of probes Oligo-B11 and Oligo-pDb12H appeared to clearly distinguish the St, J and J^S^ chromosomes in TAF46 (Figure 2a). The 42 wheat chromosomes consisted each of 12 A, 14 B, and 16 D chromosomes, including two pairs of 5BS.7BS and 5BL.7BL reciprocal translocation chromosomes, as well as the absence of chromosome 6A substituted by four chromosome 6D in TAF46 (Figure 2a,b). FISH using the probes Oligo-B11 and Oligo-pDb12H for TAF46 showed the absence of Oligo-pDb12H hybridization signals, indicating that TAF46 did not possess J^S^ chromosomes. A total six St and eight J chromosomes were observed based on the hybridization patterns of Oligo-B11 (Figure 2a,b). This is consistent with the previous reports of St-genome based GISH [3]. The Oligo-FISH painting by seven probes gave rise to distinct hybridization signals covering wheat and *Th. intermedium* chromosome arms along their entire lengths for each homoeologous group. The probe Synt1 produced strong green signals on the chromosome pairs of 1A, 1B and 1D (Figure 2c), in addition to another a pair of *Th. intermedium* group 1 chromosomes, which was defined as chromosomes 1J compared to the standard karyotype (Figure 2b,d). Similarly, the probe Synt3 generated distinct red signals on eight chromosomes including the previously identified 3A, 3B and 3D by Oligo-pSc119.2 + Oligo-pTa535 (Figure 2c), and a pair of *Th. intermedium* 3J-chromosomes (Figure 2c). Similarly, the bulked probes Synt2 + Synt5 (Figure 3e) and Synt4 + Synt6 (Figure 2g) also hybridized specifically to chromosomes of TAF46, compared to the sequential ND-FISH of Oligo-pSc119.2 + Oligo-pTA535 (Figure 2f,h). These results show that *Th. intermedium* chromosomes of TAF46 consisted of J genome chromosomes of the homoeologous groups 1, 3, 5, 7 and St genome chromosomes of groups 2, 4 and 6 (Figure 2). The karyotype for individual chromosomes of the subgenomes and linkage groups of *Th. intermedium* of TAF46 is shown in Figure 2j. Our results are consistent with the chromosome assignment of *Th. intermedium* addition lines by GISH and molecular analysis of the TAF46-derived wheat–*Thinopyrum* addition lines L1 to L7 [21].

We used a similar approach to precisely identify the *Th. intermedium* chromosomes in 78,829 (2n = 56). The ND-FISH using Oligo-pSc119.2 and Oligo-pTa535 revealed that 78,829 has complete wheat chromosomes, each of 14 A, B, and D chromosomes. The sequential ND-FISH by probes Oligo-B11 and Oligo-pDb12H revealed the *Thinopyrum* chromosomes included six St, four St-J^S^, two J-J^S^ and two J^S^ chromosomes (Figure 1 and Figure 3b). It is also interesting to observe that an Oligo-B11 signal appeared in the terminal region of 4D, indicating a small translocation between 4D and a *Th. intermedium* chromosome (Figure 3a). In comparison with ND-FISH patterns using Oligo-pSc119.2 + Oligo-pTa535 and in combination with bulked pool probes Synt1 + Synt7 (Figure 3c), Synt3+ Synt6 (Figure 3e), Synt2 + Synt5 (Figure 3g) and Synt4 (Appendix A), we constructed the karyotype of the 14 *Th. intermedium* chromosomes of 78829, as linkage groups 3St, 4St, 5St, 6J^S^-J, 7J^S^, 1St-J^S^ and 2St-J^S^, respectively. The *Th. intermedium* chromosome karyotype is shown in Figure 3j. Therefore, the bulked Oligo-based FISH painting was effective in identifying each of the wheat chromosomes or chromosome segments for particular linkage groups in the wheat–*Th. intermedium* amphiploid.

### 2.3. Constitution of Th. intermedium Chromosomes in 12 Wheat–Thinopyrum Amphiploids

By the above-mentioned strategies with ND-FISH and Oligo-FISH painting, we also determined the karyotypes of 12 wheat–*Th. intermedium* partial amphiploids TE-1502 and TE-1508, TH101-2, y70-1-4, TAI7045 and TAI7047, Zhong2, Zhong3, Zhong4, Zhong5, 78784 and 8024 (Appendix A). About 12 to 16 *Th. intermedium* chromosomes were observed in these amphiploids. The *Th. intermedium* chromosomes with their subgenome and linkage groups are shown in Figure 4 and Table 1. Among the identified partial amphiploids (Appendix A), only 20 chromosomes involving J, J^S^ and St subgenomes were observed with the absence of the 2J chromosome. The St (42.8%) and J^S^ chromosomes (35.9%) were transmitted higher than the J chromosomes (21.3%), with the highest frequency being for 7J^S^ in partial amphiploids. The linkage group 4 with 4St, 4J and 4J^S^ appeared equally transmitted in these partial amphiploids. About 32.8% of chromosomes displayed the homologous translocation between St-J^S^, J-J^S^ and St-J, while only 2 of 99 chromosomes had the non-homologous translocation of 2J^S^-5J^S^ and 3St-6St. The amphiploids TE-1502 and TAI7047 had 16 different chromosomes from *Th. intermedium*. Zhong 3, Zhong 4 and Zhong 5 displayed a close karyotype of *Th. intermedium*, except the varied number of linkage groups 2 and 4 chromosomes. Moreover, there was different number of wheat chromosomes in amphiploids. The lines TE-1502 and TE-1508 had 4J and 4J^S^ substituted for wheat chromosome 4B and Zhong 2 had four 6A chromosomes; 8024 had an absence of 7D during the transmission of the lines (Table 1).

### 2.4. Revisiting the Karyotype of Wheat–Th. intermedium Additions Z3

The wheat–*Th. intermedium* chromosome addition line Z3 contained a pair of short-satellited chromosomes with clear nucleolus regions. The ND-FISH using probes Oligo-B11 + Oligo-pDb12H, showed that the added *Th. intermedium* chromosomes were a J-St-JS translocation, and the sequential ND-FISH using probes Oligo-pSc119.2 + Oligo-pTa535 showed that these chromosomes had weak signals (Figure 5a). The Oligo-FISH painting using probes Synt1 and Synt5 was used to screen the metaphase spreads of Z3. The probe Synt1 produced strong green signals on the chromosomes 1A, 1B and 1D, and the probe Synt5 generated the red signals on chromosomes 5A, 5B and 5D, respectively, while the added *Th. intermedium* chromosomes showed the green and red signals in each arm (Figure 5b). Therefore, the Oligo-FISH painting results indicated that the added *Thinopyrum* chromosomes in Z3 were a pair of translocated chromosomes involving the short arms of group 5 and long arms of group 1.

The sequential ND-FISH with probes Oligo-5SrDNA, Oligo-pTa71, Oligo-3A1 and Oligo-pSt122 was performed to confirm the karyotype of the added *Th. intermedium* chromosomes of Z3. The signals from Oligo-5SrDNA, Oligo-pTa71 were observed in the interstitial regions of the satellite regions of *Th. intermedium* chromosomes in Z3 (Appendix A). The alien chromosome in Z3 has strong Oligo-3A1 hybridization signals on the short arm closed to centromere, and Oligo-pSt122 signals at the telomeric ends of long arms. In addition, the PCR analysis by molecular markers of group 5S (CINAU1462, CINAU1463, CINAU1472, CINAU 1489) and group 1L (TNAC1021, TNAC1042, TNAC1076, TNAC1088) produced the *Th. intermedium* specific amplification of Z3 (Figure 6). The evidence provided by molecular markers and the standard *Th. intermedium* karyotype (Figure 1) indicates that the *Th. intermedium* chromosome in Z3 was 5JS.1St-J^S^L. Therefore, the Oligo-FISH painting combined with molecular markers provide an opportunity for the precise identification of a complex chromosome rearrangement.

## 3. Discussion

Assigning a chromosome to a specific linkage group in polyploid Triticeae species with extremely large genomes traditionally relies on aneuploid analysis combined with individual chromosome recognition [22]. However, such aneuploid stocks are difficult to maintain and complete sets of aneuploids are not available for all Triticeae species [23]. Chromosome-banding techniques have long been considered as a fast, reliable, and economical means for the identification of chromosomes for wheat and related species; however, chromosome banding represents uninformative heterochromatin blocks [24,25,26,27,28,29,30,31,32,33,34,35]. Previous studies revealed that *Th. intermedium* possessed a large amount of cytogenetic polymorphism and structural modifications of chromosomes as revealed by C-banding [24]. Moreover, conventional FISH for chromosome identification was principally based on the dispersed or tandem repetitive DNA sequences, which are mostly not linkage group-specific [16,20,23,26]. Synthesizing short single-copy oligonucleotide pools has provided a new and affordable method to develop chromosome specific painting probes for FISH [27,28]. We first applied seven bulked oligos Synt1 to Synt7 as permanent resources, which enabled us to identify the homoeologous groups of chromosomes from Triticeae species fast, at a low cost and with high efficiency [13,14]. In the present study, we conducted the sequential ND-FISH probed by Oligo-B11 and Oligo-pDb12H to distinguish each of the three genomes of St, J and J^S^ in *Th. intermedium.* After that, Oligo-FISH painting by probes Synt1 to Synt7 was used to identify the seven individual linkage groups. In the present study, we first set up all the 21 pairs of *Th. intermedium* chromosomes with their precise assignment of subgenomes and linkage groups based on the Oligo-FISH painting probes and ND-FISH with subgenome-related probes (Figure 1). Our protocols can be used to karyotype the different accessions of *Th. intermedium*. The heterozygosity of each linkage group of the *Th. intermedium* was revealed (Figure 2). The results indicate that the heterozygous FISH patterns were observed in seven chromosome pairs, indicating the complexity of the karyotype of *Th. intermedium*, which is difficult to be identified by previous cytogenetic methods. Recombination among inter-genomic and non-homologous chromosome pairs was not found in the present *Th. intermedium* accession. The appropriate multiplex probes could be applied for studying chromosome constitution and chromosome variation for a large number of wild wheatgrass genetic resources.

Cauderon [31] and Cauderon et al. [32] reported the production of a wheat–*Th. intermedium* partial amphiploid TAF46 with 56 chromosomes, and it was firstly identified to have seven pairs of alien chromosomes and wheat 5B-7B reciprocal translocation by C-banding [24]. GISH using a genomic DNA from *Ps. strigosa*, Chen et al. [33] clearly determined that TAF46 has six S and eight J genome chromosomes in a wheat background, and also found that chromosome 6A was missing in an aneuploid of TAF46. In the present study, we precisely identified the individual St and J chromosomes in TAF46 and confirmed that four 6D chromosomes substituted for 6A of wheat (Figure 3). Zhong 5 has the *Thinopyrum* chromosome composition, including four S, two J^S^ and eight S-J^S^ or S-J translocation chromosomes [33]. The present study defined the linkage group of the individual *Thinopyrum* chromosome in Zhong 5, which is consistent with the previous ND-FISH results from the Zhong 5-derived addition lines [24]. The Robertsonian translocation between 1St and 1J^S^, 2J^S^ and 2St, as well as two small translocations, occurred between 4D and *Th. intermedium* chromosomes in 78,829 (Figure 3). In the present study, the Oligo-FISH painting probes facilitated the study of inter-species chromosome homologous relationships and visualized non-homologous chromosomal rearrangements in some wheat-*Thinopyrum* derivatives. Our results provide a precise recognition of individual *Th. intermedium* chromosome and their rearrangement of the 13 wheat–*Th. intermedium* partial amphiploids, which will be helpful for the subsequent transfer of *Thinopyrum* chromatin from amphiploid to wheat. The high transmission rate of chromosome 7J^S^ in the identified amphiploids may be partially due to the presence of novel disease- and insect-resistance genes [4,34,35].

The wheat–*Th. intermedium* partial amphiploid contained stable karyotypes, since the homoeologous group of three subgenome chromosomes have high complementary [4]. However, their chromosome structures were easily rearranged following their hybridization with wheat [5]. The wheat–*Th. intermedium* introgression lines have been developed to transfer novel gene(s) from the diversified gene-pool of *Thinopyrum* to wheat [5,8]. Tang et al. [34] revealed that the addition line Z3 derived from Zhong 5 contained a pair of *Th. Intermedium*-derived small-satellite chromosomes by GISH. Hu et al. [36] revealed that the C-banding patterns and the FISH signals by pTa71 hybridization of 1St#2 chromosomes in AS1677 differed from those in Z3. In the present study, the ND-FISH with multiple probes and the Oligo-FISH painting methods indicated that the added *Th. intermedium* chromosome in Z3 is a 5JS.1St-J^S^L translocation (Figure 5). The *Th. intermedium* chromosomes in Z3 are derived from the chromosome set in the parent Zhong 5 (Figure 4). Our previous study identified that the line Hy37, which originated from Zhong 5, contained a 5JS.3StS chromosome [24], indicating frequent chromosome modifications after the wheat–*Th. intermedium* amphiploid was crossed to wheat. Therefore, the oligo-painting probe-based FISH proved to be a useful tool for visualizing the occurrence of chromosome rearrangements during early and later generations of wheat-alien transfer subsequent to wide hybridization and chromosome engineering.

High molecular weight glutenin subunits (HMW-GSs) are encoded by the *Glu-1* loci located on the long arm of group-1 chromosomes of Triticeae species [37,38]. Several genes for HMW-GS have been identified in the subgenomes of *Th. intermedium* [39]. Niu et al. [40] systematically characterized the HMW-GS composition of several wheat–*Th. intermedium* partial amphiploids and derivatives, and found that TAF46 and Zhong 2 expressed the *Thinopyrum*-specific HMW-GS. Hu et al. [36] confirmed that the *Th. intermedium* ssp. *Trichophorum*-derived partial amphiploid and substitution line AS1677 expressed the 1St-specific HMW-GS. The present study found that both TAF46 and Zhong 2 have the 1J chromosome, while other amphiploids contained the 1StS.1J^S^L-rearranged chromosome (Figure 4). The line Z3 with 5JS.1J^S^L did not express *Thinopyrum*-specific HMW-glutenin subunits [40]. It is worthwhile to investigate HMW-GS expression in newly developed *Th. intermedium* 1J^S^ substitution lines [41,42], and also to reveal the structure of the 1J^S^L in Z3 and determine the absence of the expression of HMW-GS. The HMW-GS gene expansion and expression will be studied in detail to reveal the mechanism of the changes during allo-polyploidization and introgression [43]. The ongoing precise genome sequencing of *Th. intermedium* species at a chromosomal level and the re-sequencing of wheat–*Th. intermedium* derivatives will be essential to dissect the complex evolutionary history of Triticeae species [43,44,45,46].

## 4. Materials and Methods

### 4.1. Plant Materials

*Th. intermedium* PI440043 (StJ^S^J, 2n = 6x = 42) was obtained from the National Small Grains Collection at Aberdeen, Idaho, USA. Wheat–*Th. intermedium* partial amphiploid TAF46 [21] was kindly provided by Dr. Bernd Friebe, Wheat Genetics Resource Center, Kansas State University, USA. Wheat lines Chinese Spring (CS); wheat–*Th. intermedium* ssp. *trichophorum* partial amphiploid TE-1508, TE-1502, TH101-2 and y70-1-4; wheat–*Th. intermedium* addition lines Z3 [22] and Hy37 [18] are maintained by the Center for Informational Biology, School of Life Science and Technology, University of Electronic Science and Technology of China. The wheat–*Th. intermedium* partial amphiploids TAI7045 and TAI7047, Zhong2, Zhong3, Zhong4, Zhong5, 78784, 78829 and 8024 were obtained from Dr. Zhijian Chang, Shanxi Academy of Agricultural Sciences, China.

### 4.2. Probe Preparation

Seven oligo-FISH pools of probes Synt1 to Synt7 corresponded to each of the seven Triticeae homoeologous groups, respectively [13]. These oligo probes were selected from the single-copy sequences derived from 1H to 7H barley chromosomes with a 96% sequence homology with wheat linkage groups 1 to 7 chromosomes, which enabled us to distinguish the chromosomes for specific linkage groups. The bulked oligo libraries Synt1 to Synt7 were synthesized by MYcroarray (Ann Arbor, MI, USA). Probe preparation from the synthesized oligo library was performed as described by Han et al. [29]. The tandem repeat-based oligo-nucleotide probes for ND-FISH are listed in Appendix A. Labeled Oligonucleotide probes were synthesized by Shanghai Invitrogen Biotechnology Co. Ltd. (Shanghai, China). The labeling and preparation of bulk painting oligos was conducted following the description of Li and Yang [14].

### 4.3. Fluorescence In Situ Hybridization

Root tips from germinated seeds were collected and treated with nitrous oxide followed by enzyme digestion, using the procedure of Han et al. [47]. The synthetic oligonucleotides were either 5′ end-labelled with 6-carboxyfluorescein (6-FAM) for green or 6-carboxytetramethylrhodamine (Tamra) for red signals. The protocol of non-denaturing FISH (ND-FISH) by the synthesized probes was described by Fu et al. [19]. After the oligo-based FISH, the sequential FISH with bulk painting with oligos was conducted following the description by Li and Yang [14]. Photomicrographs of FISH chromosomes were taken with an Olympus BX-53 microscope equipped with a DP-70 CCD camera.

### 4.4. Molecular Marker Analysis

The PLUG markers [48] and CINAU primers [49] for location on specific chromosomes were obtained by searching the database of Wheat Genome Assembly ref. v1.0. The PCR protocol used the 8% PAGE gel separation was as described by Hu et al. [36].

## 5. Conclusions

The present oligo-bulked pool FISH visualized homoeologous regions directly on Triticeae chromosomes in a simple and fast experimental procedure. Our present comparative genomic-based oligo-painting FISH studies provide new insights into the evolution of the *Th. intermedium* genomes. The protocol of FISH with multiple types of oligos has great potential for the high-throughput karyotyping of wheat–*Th. intermedium* introgression lines for effective wheat breeding by chromosome manipulation.

## Figures and Tables

**Figure 1 plants-11-02109-f001:**
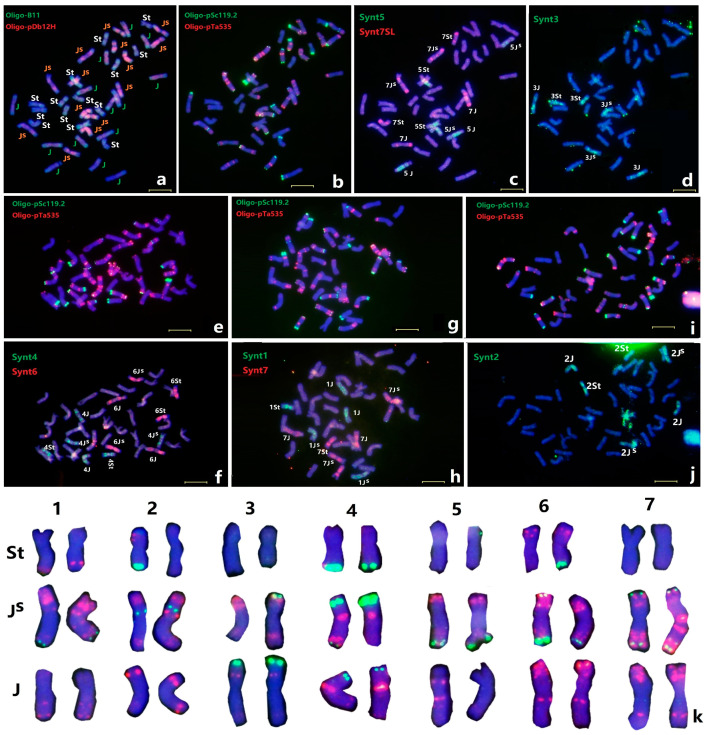
Sequential FISH for *Th. intermedium* PI440043 with probes Oligo-B11 + Oligo-pDb12H (**a**), Oligo-pSc119.2 + Oligo-pTa535 (**b**,**e**,**g**,**i**), Synt5 + Synt7SL (**c**), Synt3 (**d**), Synt4 + Synt6 (**f**), Synt1 and Synt7 (**h**) and Synt2 (**j**), respectively. The chromosomes were counterstained with DAPI (blue). (**k**) Karyotypes of *Thinopyrum intermedium* PI440043 with homoeologous and subgenome assignment were subjected to sequential ND-FISH with Oligo-pSc119.2 + Oligo-pTa535. Bars, 10 μm.

**Figure 2 plants-11-02109-f002:**
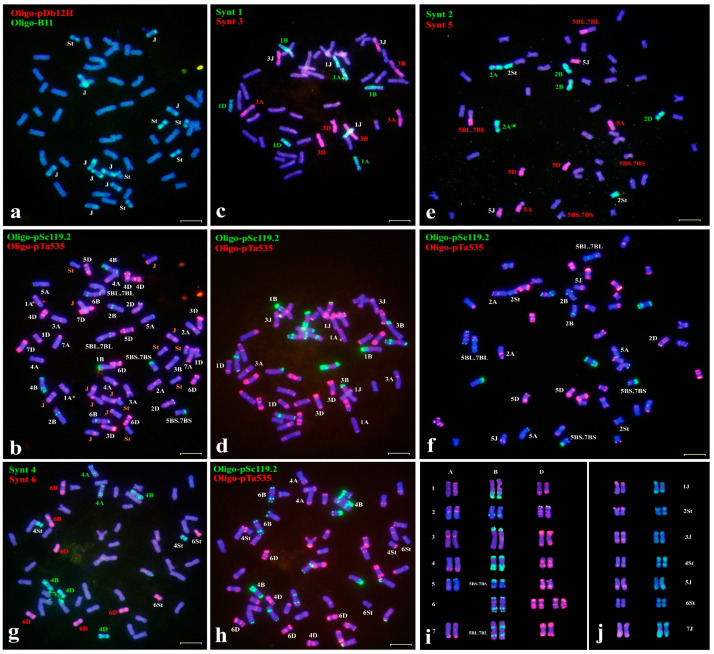
Sequential FISH probes Oligo-B11 + Oligo-pDb12H (**a**), Oligo-pSc119.2 +Oligo-pTa535 (**b**,**d**,**f**,**h**) and Oligo-FISH painting using specific bulked oligo probes Synt1+ Synt3 (**c**), Synt2 + Synt5 (**e**) and Synt4 + Synt6 (**g**) for wheat–*Th. intermedium* partial amphiploid TAF46, respectively. Karyotype of wheat chromosomes (**i**) and *Th. intermedium* (**j**) are shown. Chromosomes were counterstained with DAPI (blue). Bars, 10 μm.

**Figure 3 plants-11-02109-f003:**
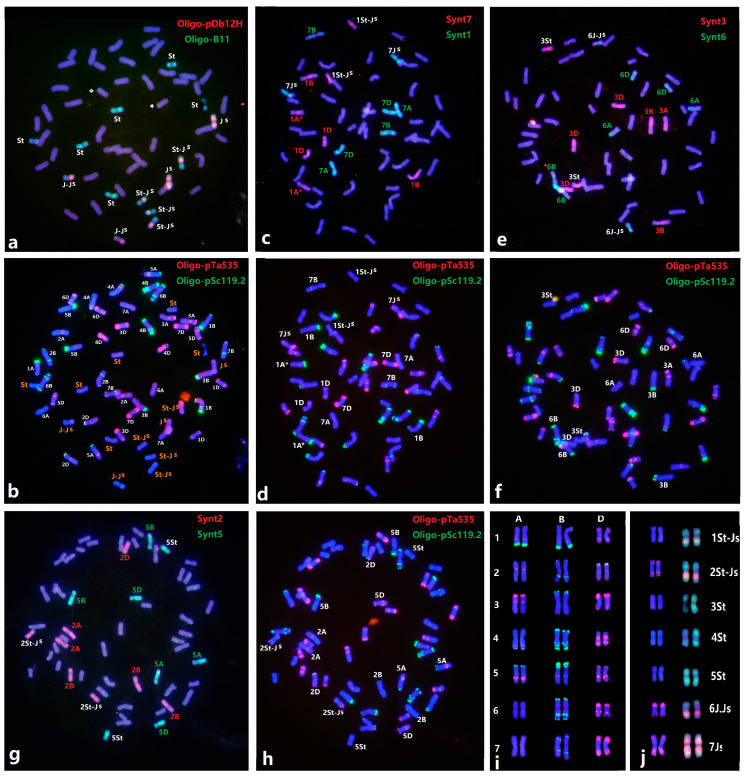
Sequential FISH probes Oligo-B11 + Oligo-pDb12H (**a**), Oligo-pSc119.2 + Oligo-pTa535 (**b**,**d**,**f**,**h**) and Oligo painting using specific bulked oligo probes Synt1+ Synt7 (**c**), Synt3 + Synt6 (**e**) and Synt2 + Synt5 (**g**) for wheat–*Th. intermedium* partial amphiploid 78829, respectively. Karyotypes of wheat chromosomes (**i**) and *Th. intermedium* (**j**) are shown. Chromosomes were counterstained with DAPI (blue). Arrows showed the wheat–*Thinopyrum* small translocation chromosomes.

**Figure 4 plants-11-02109-f004:**
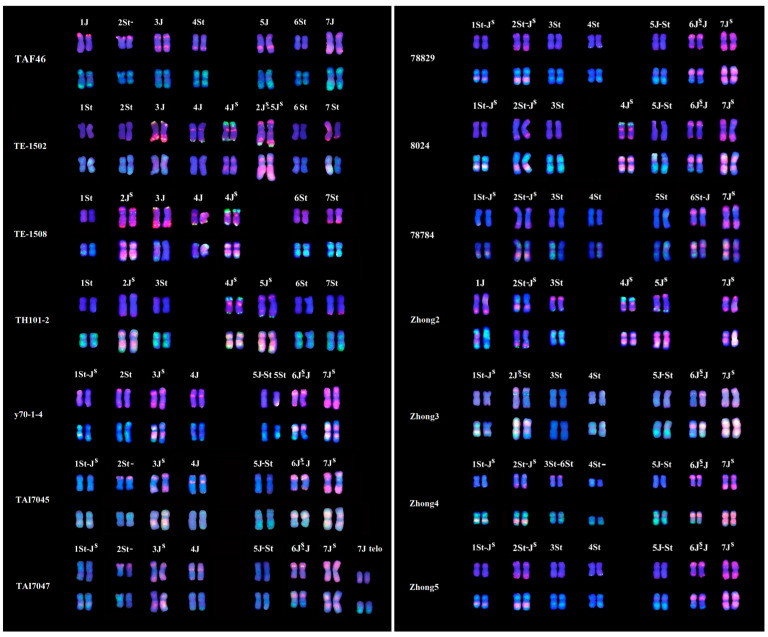
Karyogram of *Th. intermedium* chromosomes in wheat–*Th. intermedium* partial amphiploids based on FISH signals of Oligo-pSc119.2 + Oligo-pTa535 (up), and Oligo-B11 + Oligo-pDb12H (bottom). The homoeologous groups 1 to 7 of the *Th. intermedium* were determined by Oligo-painting probes Synt1 to Synt7. The FISH plates of various partial amphiploids are shown in Table 1 and Appendix A.

**Figure 5 plants-11-02109-f005:**
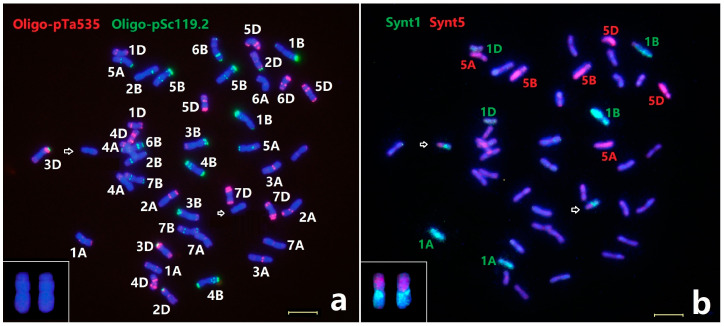
Sequential ND-FISH and Oligo-FISH painting of the wheat–*Th. intermedium* addition line Z3. (**a**) Probes Oligo-pSc119.2 (green) + Oligo-pTa535 (red) and (**b**) Synt1 (green) + Synt5 (red) are shown. The added *Thinopyrum* chromosomes are shown by arrows and the cut-and-paste chromosomes at the bottom left. Bars, 10 μm.

**Figure 6 plants-11-02109-f006:**
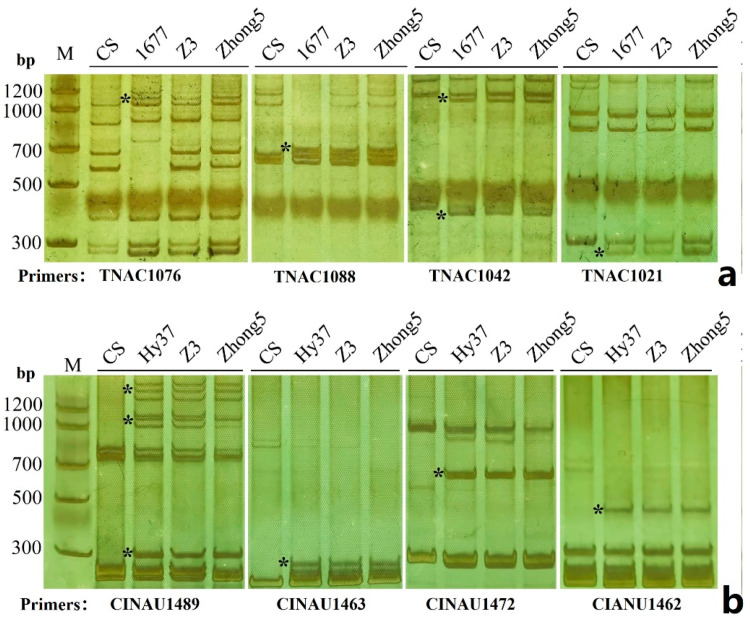
PCR analysis using TNAC and CINAU primers of the short arms of group 1 (**a**) and group 5 (**b**). Materials, CS (Chinese Spring); Hy37 (5JS.3StS translocation line); AS1677 (1St-1D substitution); Z3, Zhong5. The * indicate *Th. Intermedium*-specific bands.

**Table 1 plants-11-02109-t001:** The *Thinopyrum* chromosome constitution of the wheat–*Th. intermedium* partial amphiploids.

Materials	Chromosome Number	Chromosome Constitution	Figure
Zhong2	2n = 56, four 6A	1J, 2St-J^S^, 3St, 4J^S^, 5J^S^, 7J^S^	Appendix A
Zhong3	2n = 56	1St-J^S^, 2J^S^-St, 3St, 4St, 5J-St, 6J^S^-J, 7J^S^	Appendix A
Zhong4	2n = 56	1St-J^S^, 2St-J^S^, 3St-6St, 4St-, 5J-St, 6J^S^-J, 7J^S^	Appendix A
Zhong5	2n = 56	1St-J^S^, 2St-J^S^, 3St, 4St, 5J-St, 6J^S^-J, 7J^S^	Appendix A
TAI7045	2n = 56	1St-J^S^, 2St, 3J^S^, 4J, 5J-St, 6J^S^-J, 7J^S^	Appendix A
y70-1-4	2n = 56	1St-J^S^, 2St, 3J^S^, 4J, 5J-St (1), 5St (1), 6J^S^-J, 7J^S^	Appendix A
78784	2n = 56	1St-J^S^, 2St-J^S^, 3St, 4St, 5St, 6J^S^-J, 7J^S^	Appendix A
TAI7047	2n = 56 + t	1St-J^S^, 2St, 3J^S^, 4J, 5J-St, 6J^S^-J, 7J^S^, 7J telo	Appendix A
78829	2n = 56	1St-J^S^, 2St-J^S^, 3St, 4St, 5J-St, 6J^S^-J, 7J^S^	Figure 3
TE-1502	2n = 56, absence of 4B	1St, 2St, 3J, 4J, 4J^S^, 2J^S^-5J^S^, 6St, 7St	Appendix A
TE-1508	2n = 56	1St, 2J^S^, 3J, 4J, 4J^S^, 6St, 7St	Appendix A
TH101-2	2n = 56	1St, 2J^S^, 3St, 4J^S^, 5J^S^, 6St, 7St	Appendix A
8024	2n = 54, absence of 7D	1St-J^S^, 2St-J^S^, 3St, 4J^S^, 5J-St, 6J^S^-J, 7J^S^	Appendix A
TAF46	2n = 56	1J, 2St-, 3J, 4St, 5J, 6St, 7J	Figure 2

## Data Availability

Data are available upon request. Data are contained within the article.

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
