# Peer review of "Precise Identification of Chromosome Constitution and Rearrangements in Wheat–Thinopyrum intermedium Derivatives by ND-FISH and Oligo-FISH Painting"

_plants, 2022, doi:10.3390/plants11162109_

Round 1

Reviewer 1 Report

The title of the article should be changed as suggested in the revised manuscript attached as there is more information on the genomic and chromosomal constitution of Th. intermedium, wheat-Th. intermedium partial amphiploids and derivative lines than the limited chromosomal rearrangement. The introduction should be elaborated to show readers' interest in using the information for introgression of useful variability from Th. intermedium indicating what type of variability has already been introgressed and utilized. The application of various FISH probes for the identification of genomes and chromosomes should be described briefly as the readers may not have to consult the referred articles for interpreting and using the results. The figure legends should be extensively revised giving the details of the species, amphiploid, partial amphiploids and derivatives, and the probes used for sequential FISH. Full detail of the material and probe used should be tabulated along with references  It is not clear why the modified Js chromosomes are identified with Oligo-pDb124 probe while the original St and J chromosomes are not clearly identified by a joint probe Olig-B11. It is interesting to know that the partial amphiploids are stabilized at 2n=56 irrespective of their genomic constitution. It should be explained in the discussion section. There are seven homeologous linkage groups in Triticeae and it has been possible to identify the subgenomic chromosomes using various sequential FISH probes hence the word homeologous should be invariably used along the linkage group throughout the text as suggested. A number of interpretation and grammatical mistakes have been corrected which should be taken into consideration while revising the manuscript. Some sentences have been highlighted which are ambiguous and should be clarified.  

Author Response

  1. The title of the article should be changed as suggested in the revised manuscript attached as there is more information on the genomic and chromosomal constitution of Th. intermedium, wheat-Th. intermedium partial amphiploids and derivative lines than the limited chromosomal rearrangement.

Response: We agree to revise the title as suggested.

  1. The introduction should be elaborated to show readers' interest in using the information for introgression of useful variability from Th. intermedium indicating what type of variability has already been introgressed and utilized.

Response: We agree the comments. Several disease resistance genes for the rusts and powdery mildew resistance, and new glutenin genes for quality enhancement from Th. intermedium have been introduced to wheat.

  1. The application of various FISH probes for the identification of genomes and chromosomes should be described briefly as the readers may not have to consult the referred articles for interpreting and using the results.

Response: We agree to revise the part as suggested. The probes and the references are listed in Table S1.

  1. The figure legends should be extensively revised giving the details of the species, amphiploid, partial amphiploids and derivatives, and the probes used for sequential FISH. Full detail of the material and probe used should be tabulated along with references.

Response: We agree to revise the Figure legends and add the detail information in Table S1.

  1. It is not clear why the modified Js chromosomes are identified with Oligo-pDb12H probe while the original St and J chromosomes are not clearly identified by a joint probe Olig-B11.

Response: The probe pDb12H is a Sabrina type of LTR isolated by Yang et al. (2006). pDb12H for FISH and Oligo-pDb12H for ND-FISH has V and Js genome specific, which is identical for GISH studies of Thinopyrum intermedium (Liu et al. 2009; Xi et al. 2019).

Yang, Z. J., Liu, C., Feng, J., Li, G. R., Zhou, J. P. Deng, K. J. and Ren, Z. L. 2006. Studies on genomic relationship and specific marker of Dasypyrum breviaristatum in Triticeae. Hereditas, 143: 47-54.

Liu C, Yang ZJ, Jia JQ, Li GR, Zhou JP, Ren ZL. 2009. Genomic distribution of a Long Terminal Repeat (LTR) Sabrina-like retrotransposon in Triticeae species. Cereal Research Communications 37(3): 363–372.

Xi W, Tang Z, Tang S, Yang Z, Luo J, Fu S. 2019. New ND-FISH-positive Oligo probes for identifying Thinopyrum chromosomes in wheat backgrounds. Int J Mol Sci., 20, 2031.

Yu Z, Wang H, Xu Y, Li Y, Lang T, Yang Z, Li G. 2019. Characterization of chromosomal rearrangement in new wheat - Thinopyrum intermedium addition lines carrying Thinopyrum-specific grain hardness genes. Agronomy, 9, 18.

  1. It is interesting to know that the partial amphiploids are stabilized at 2n=56 irrespective of their genomic constitution. It should be explained in the discussion section.

Response: The wheat-Thinopyrum intermedium partial amphiploid became relatively stable, since the chromosomes among the same homoeologous group of three sub-genomes of St-, J- and Js have high complementary in wheat background.

  1. There are seven homoeologous linkage groups in Triticeae and it has been possible to identify the subgenomic chromosomes using various sequential FISH probes hence the word homoeologous should be invariably used along the linkage group throughout the text as suggested.

Response: We agree to change the description of linkage to homoeologous as suggested.

  1. A number of interpretation and grammatical mistakes have been corrected which should be taken into consideration while revising the manuscript. Some sentences have been highlighted which are ambiguous and should be clarified.

Response: Thank you for the correction for our manuscript. The Dr. Ian Dundas of University of Adelaide also edited the grammatical errors in the main text of the revised version.

Reviewer 2 Report

Thinopyrum intermedium is an invaluable genetic source in wheat improvement. In this manuscript, authors established standard karyotype of individual Th. intermedium chromosomes and identified individual chromosomes in wheat-Th. intermedium partial amphiploid by comprehensive FISH and Oligo-FISH painting. It is very interesting that this study may help to uncover the chromosome evolution and promote the breeding in wheat. It was well written. Especially its results were well summarized and presented. However, there are still some minor concerns with me.

1) In your topic, you said that you can precisely identify rearrangements in wheat-Thinopyrum intermedium. I do not think that it is supported by your results. You can only tell the rough location of rearrangements on chromosomes with FISH or molecular markers. Suggesting a minor

modification to it.

2) line 60, is it “shown” or shew?

3) Many references require to be formatted with the same doi. And some reference numbers are in blue. Please modify them with the same format.

4) Fig. S1., bars are required. 

Author Response

1. In your topic, you said that you can precisely identify rearrangements in wheat-Thinopyrum I do not think that it is supported by your results. You can only tell the rough location of rearrangements on chromosomes with FISH or molecular markers. Suggesting a minor modification to it.

Response: We agree to change the description. The Oligo-B11 and Oligo-pDb12H was used to identify the Thinopyrum chromatin in wheat background. Most of the non-homoeologous rearrangement identified within the Th. intermedium chromosomes.  

2. line 60, is it “shown” or shew?

Response: I agree to change

3. Many references require to be formatted with the same doi. And some reference numbers are in blue. Please modify them with the same format.

Response: I agree to change the references as suggested.

4. Fig,S1., bars are required. 

Response: I agree to add the bars.